# Quantification of Boron Compound Concentration for BNCT Using Positron Emission Tomography

**DOI:** 10.3390/cells9092084

**Published:** 2020-09-12

**Authors:** Marcin Balcerzyk, Manuel De-Miguel, Carlos Guerrero, Begoña Fernandez

**Affiliations:** 1Centro Nacional de Aceleradores (Universidad de Sevilla—CSIC—Junta de Andalucía), C/Thomas Alva Edison 7, 41092 Sevilla, Spain; cguerrero4@us.es (C.G.); bfernand@us.es (B.F.); 2Departamento de Fisiología Médica y Biofísica, Facultad de Medicina, Universidad de Sevilla, Avda. Sánchez-Pizjuán s/n, 41009 Sevilla, Spain; 3Departamento de Citología e Histología Normal y Patológica, Universidad de Sevilla, Avda. Sánchez-Pizjuán s/n, 41009 Sevilla, Spain; mmiguel@us.es

**Keywords:** boron neutron capture therapy, positron emission tomography, thyroid, sodium-iodine symporter

## Abstract

**Background**: Boron neutron capture therapy requires a 2 mM ^10^B concentration in the tumor. The well-known BNCT patient treatment method using boronophenylalanine (BPA) as a boron-carrying agent utilizes [^18^F]fluoroBPA ([^18F^]FBPA) as an agent to qualify for treatment. Precisely, [^18^F]FBPA must have at least a 3:1 tumor to background tissue ratio to qualify the patient for BNCT treatment. Normal, hyperplasia, and cancer thyroids capture iodine and several other large ions, including BF_4_^−^, through a sodium-iodine symporter (NIS) expressed on the cell surface in normal conditions. In cancer, NIS is also expressed within the thyroid cell and is not functional. **Methods**: To visualize the thyroids and NIS, we have used a [^18^F]NaBF_4_ positron emission tomography (PET) tracer. It was injected into the tail veins of rats. The [^18^F]NaBF_4_ PET tracer was produced from NaBF_4_ by the isotopic exchange of natural ^19^F with radioactive ^18^F. Rats were subject to hyperplasia and tumor-inducing treatment. The NIS in thyroids was visualized by immunofluorescence staining. The boron concentration was calculated from Standard Uptake Values (SUV) in the PET/CT images and from the production data. Results: 41 MBq, 0.64 pmol of [^18^F]NaBF_4_ PET tracer that contained 0.351 mM, 53 nmol of NaBF_4_ was injected into the tail vein. After 17 min, the peak activity in the thyroid reached 2.3 MBq/mL (9 SUV_max_). The ^nat^B concentration in the thyroid with hyperplasia reached 381 nM. **Conclusions**: Such an incorporation would require an additional 110 mg/kg dose of [^10^B]NaBF_4_ to reach the necessary 2 mM ^10^B concentration in the tumor. For future BNCT treatments of thyroid cancer, contrary to the ^131^I used now, there is no post-treatment radioactive decay, the patient can be immediately discharged from hospital, and there is no six-month moratorium for pregnancy. This method can be used for BNCT treatment compounds of the type R-BF_n_, where 1 <= *n* <= 3, labeled with ^18^F relatively easily, as in our example. A patient may undergo injection of a mixture of nonradioactive R-BF_n_ to reach the necessary ^10^B concentration for BNCT treatment in the tumor together, with [^18^F]R-BF_n_ for boron mapping.

## 1. Introduction

Radiation treatment is one of the widely accepted and used forms of cancer treatment. Currently, irradiation with energetic gamma is a mainstream technology. As a second, the irradiation with protons and ^12^C ions are already introduced in more than 80 centers around the world. The main drawback of these types of irradiation is that the beam energy is deposited on the track before reaching the tumor. For proton and ^12^C irradiation, this effect is much lower but still significant. In boron neutron capture therapy (BNCT), this effect is further reduced, as the target is defined by the presence of ^10^B in the injected drug and its distribution. 

BNCT is experiencing now a revival due to the introduction of accelerator-based epithermal neutron sources, which can be installed in hospitals [1]. The explored reaction in the accelerator is ^7^Li(p,n)^7^Be [2]. It is subject to extensive development, as, for therapeutically useful epithermal neutron fluences, up to 50 kW of the energy is dissipated. The next challenge is the selection of a boron compound that targets preferentially tumor cells. Up to now, the best BNCT results were achieved using boronophenylalanine (BPA) produced using enriched boron ^10^B [3]. The patient is qualified for BNCT using BPA if a previous PET study using [^18^F]fluoro-BPA ([^18^F]FBPA, [4]) shows a tumor to background capture ratio of at least 3:1. There is active research in search of new boron compounds with higher tumor to background ratios and slow enough clearances so that all boron treatments can be done in one hour with still fluence-limited epithermal accelerator sources. 

Prompted by that need, we have examined the possibility of the use of NaBF_4_ and a PET tracer version of this compound [^18^F]NaBF_4_ [5] for their use in BNCT. NaBF_4_ readily ionizes in water to BF_4_^−^, which is captured by a sodium-iodine symporter (NIS) in thyroids and the pituitary gland [6]

NIS-containing cells in thyroids constitute only part of the gland; therefore, the incorporated boron has a higher local concentration. NaBF_4_ and [^18^F]NaBF_4_ are administered together in the PET study injection in known proportions, allowing determination of the boron concentration in the tumor for BNCT treatment in the theranostic approach. [^18^F]NaBF_4_ is reaching a maximum concentration in the tumor 17 min after injection in the rat and maintains 70% of activity until 90 min post-injection. This allows the using of NaBF_4_ for thyroid BNCT treatment, which, thus, shows some advantages over traditionally used radioactive ^131^I. As [^18^F]NaBF_4_ is produced from NaBF_4_ by the simple isotopic exchange at moderate temperatures, the boron concentration monitoring can be easily extended to R-BF_n_-type compounds, which are more specific for tumors. 

## 2. Materials and Methods

### 2.1. Animals

Female Wistar rats, aged 16–17 weeks old and weighing 200–220 g at the study starting moment, were kept in pathogen-free cages at the animal facilities of the School of Medicine (University of Seville, Seville, Spain) under controlled laboratory conditions (temperature 20 ± 2 °C, humidity 40–50%, lighting regimen of 12 light/12 dark hours). Animals were fed a normal laboratory diet (Teklad diet, Envigo, Cambridgeshire, UK) and fresh tap water ad libitum. The study was approved by the Ethical Committee of the Faculty of Medicine, University of Seville. Animal handling was conducted following the Guide for the Care and Use of Laboratory Animals [7]. Rats were divided into two equal groups: the control group and the treated group. Animals from the treated group received 1% potassium perchlorate (*w*/*v*) in the drinking water, ad libitum, for 18 months [8]. All experiments were carried with the approval of the ethics committee of the University of Seville.

### 2.2. [18. F]NaBF_4_ PET Tracer Production

For [^18^F]NaBF_4_ PET tracer production, we followed the method described in reference [5]. Briefly, [^18^F]NaBF_4_ was produced by an isotopic exchange of natural ^19^F from NaBF_4_, with ^18^F produced in a 18-MeV IBA Molecular cyclotron (IBA, Louvain-la-Neuve, Belgium) in 1.5 M HCl solution heated for 10 min in 120 °C. The final compound was purified in an alumina cartridge. Quality control was done using HPLC. The radiotracer was prepared for injection in saline.

### 2.3. Histologic Study and NIS Immunofluorescence Staining

Thyroid glands were extracted and fixed in 10% neutral-buffered formalin, embedded in paraffin using the standard procedure, sectioned at 4–5-µm thickness, and mounted on silane-coated glass slides. Tissue sections were stained with hematoxylin and eosin (H&E) for histopathological analysis and the selection of thyroid tissue with a characteristic appearance to perform immunofluorescence (IF) for NIS.

For IF staining, tissue sections were dewaxed in xylene and hydrated through graded alcohols. Then, they were incubated with the primary antibodies (anti-NIS, a-r139p, 1:5000 dilution) overnight at 4 °C. Afterward, slides were washed in PBS and incubated for 30 min at room temperature with respective secondary antibodies labeled with Cy-3 (red) (1:200 dilution, Jackson Immuno Research Laboratories, Newmarket, UK). DAPI was added for nuclei counterstaining. Finally, after PBS washing, slides were mounted in 90% glycerol, 2% n-propyl gallate (Sigma, St. Louis, MO, USA) and observed under a fluorescence microscope (Olympus BX50, Olympus Corporation, Tokio, Japan). Controls for immunoreaction specificity were performed by omitting the primary antibody step of the technique.

### 2.4. PET Imaging and Image Processing

PET imaging was carried out in a Mosaic microPET scanner (Philips, Andover, MA, USA) followed by a CT scan in NanoCT (Bioscan/Mediso, Budapest, Hungary). Briefly, animals were anesthetized with isoflurane. A catheter was placed in the tail vein. A radiotracer (41 MBq) was injected with a maximum limit of 5 mL/kg in a bolus injection. In dynamic studies, this was done already in the PET scanner on a heated animal bed (Équipement Vétérinaire MINERVE, Esternay, France). For static studies, animals were awakened and let up for 45-min radiotracer uptake. Then, the animal was anesthetized again and placed in the heated bed of the PET scanner. The dynamic scans were of 90 min centered in the thyroids; the static ones were three beds of 10 cm longitudinal lengths starting from the head towards the tail. Each bed position was scanned for 15 min. CT scans were acquired in 45 kVp with 177 µA at 500 ms exposition and 240 projections per rotation.

PET images were reconstructed using an iterative method with decay, scatter, and random corrections without attenuation correction. The images had a uniform 1 mm pixel size. CT images were reconstructed using the filtered back-projection method with an exact cone-beam and with a Shepp Logan 98% filter, giving a 0.2 mm uniform pixel size. PET images were fused to CT images in PMOD 3.2 (PMOD Technologies LLC, Zurich, Switzerland, now part of Bruker’s Preclinical Imaging Division, Billerica, MA, USA). No partial volume correction was used. The volume of interest (VOI) for thyroids were delineated at 50% of the maximum uptake.

The radiotracer is stable in vivo [5] and shows no capture in bones. The kinetic modeling of the tracer in VOIs was done using two tissue compartment models shown in Equation (1). *C*_p_ is the activity concentration of the tracer in the plasma, *C*_1_ and *C*_2_ are concentrations in two compartments, *k* are the kinetic constants, and *t* is time. The kinetic modeling was done in the PMOD module PKIN using full blood as the input function, with linear interpolation between experimental points taken from the dynamic image of the left ventricle of the heart.



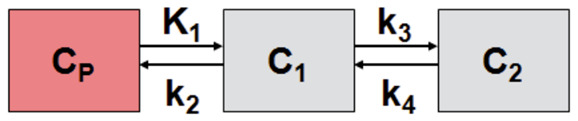



(1)$$\begin{array}{c} \frac{{\mathrm{dC}}_{2}\left(t\right)}{\mathrm{dt}}=k_{3}C_{1}\left(t\right)-k_{4}C_{2}\left(t\right)\\ \frac{{\mathrm{dC}}_{1}\left(t\right)}{\mathrm{dt}}=K_{1}C_{p}\left(t\right)-\left(k_{2}\;+{\;k}_{3}\right)C_{1}\left(t\right)\;+{\;k}_{4}C_{2}\left(t\right) \end{array}$$

### 2.5. HISPANoS as Epithermal Neutrons Irradiation Facility for BNCT

HISPANoS, from Hispalis Neutron Source (http://www.cna.us.es/HISPANOS/), is the first accelerator-based neutron source (pulsed and continuous wave) in Spain located at the 3 MV tandem accelerator at the National Accelerator Centre (CNA, Seville) [9,10]. At HISPANoS, fast, epithermal, and thermal neutrons can be delivered through different nuclear reactions [2,11,12]. In the case of the epithermal beams of interest in this work, the proton beam and beam-shaping assembly (BSA) configurations from Fantidis et al. [13] have been considered: a proton beam of 2.5 MeV, in our case, with a current of 10 µA would hit a water-cooled thick lithium target, producing, in the absence of the BSA, ~3·10^7^ n/cm^2^/s at 10 cm. This is about 30 times lower a value than the reference value of 10^9^ n/cm^2^/s considered by the International Atomic Energy Agency (IAEA) [14], hence not sufficient for clinical BNCT but enough for tests and some specific cases, with the particular feature that the neutron beam is adjacent to both the radiopharmacy laboratory and the PET scanner.

### 2.6. Calculation of Boron Concentration

As injection solution of the [^18^F]NaBF_4_ tracer contains the majority of unlabeled NaBF_4_; from the NaBF_4_ molar concentration in the synthesis solution, we could know the labeled to unlabeled tracer compound concentration ratio at the beginning of the PET study. As PET images are corrected for ^18^F decay, the tracer molar concentration (not activity concentration) gave us the unlabeled NaBF_4_ concentration and, hence, the boron concentration in the studied tissue. The boron concentration is known only with the spatial resolution of the PET image, which in the Mosaic scanner case is about 2.3 mm. The microscopic details of the boron concentration were inferred from NIS immunofluorescence, assuming that the BF_4_^−^ concentration is only significant in NIS-containing cells.

Briefly, at the radiotracer production stage, the mass of NaBF_4_, Milli-Q water, and activity and volume of the [^18^F]F^-^ Milli-Q water solution arriving from the cyclotron are known. Correcting for ^18^F decay during the production and time needed for injection, with the known volume and activity of the produced and injected [^18^F]NaBF_4_ tracer, we can calculate the amount of NaBF_4_ left from the production stage in the tissue from the activity concentration expressed initially as maximum local value of standard uptake value (SUV_max_), then recalculated to kBq/mL and, then, to the molar concentration of ^18^F at the beginning of the PET study.

## 3. Results

### 3.1. [^18^F]NaBF_4_ Radiotracer Production

We have produced a typical batch of 4 GBq of [^18^F]NaBF_4_ tracer in 2 mL of saline. The [^18^F]NaBF_4_ tracer was of 98% radiochemical purity. The labeling efficiency was 36% corrected for decay. The synthesis was completed in 40 min. The pH of the solution for injection was 6–7. One milligram of NaBF_4_ was used, and the volume of water used in the synthesis was 25.9 mL, so the concentration of unlabeled NaBF_4_ was 0.351 mM in the final solution of 2 mL ready for injection. The concentration of the [^18^F]NaBF_4_ tracer was 33.9 nM in the injectable volume at the end of synthesis (EOS), as about 4.3 GBq of [^18^F]NaBF_4_ tracer was produced. As both unlabeled and labeled NaBF_4_ was used and injected into an animal, from the proportions in the solution at the EOS, we can follow the boron concentration monitoring the ^18^F activity concentration in the PET; hence, the ^18^F and boron concentrations.

### 3.2. IF Staining of NIS in Thyroid Tissue

After NIS IF staining, it can be observed that the NIS expression is higher in both thyroid lesions than in the control of normal thyroid tissue, especially in tumor thyroid samples. The IF and H&E staining of the thyroid tissue in a control animal, an animal with thyroid hyperplasia, and thyroid tumor are shown in Figure 1.

The normal thyroid shown in Figure 1a has a pronounced colloid volume—visible as white centers surrounded by nuclei and NIS. The colloid constitutes about one-third of the thyroid volume: 2.5 µL of the 7.5 µL total thyroid volume reported for a rat for both lobes [14]. The follicles surround the colloid as the outer shell, which is about one-seventh of the radius of the colloid (14%), which can be seen in Figure 1a. For hyperplasia, it is about 25% (Figure 1b), and for tumors, it is practically full volume. The fraction of the thyroid volume occupied by follicles is, therefore, about 30% for healthy tissue, 50% for hyperplasia, and practically 100% for tumors.

### 3.3. [^18^F]NaBF_4_ PET Radiotracer Uptake in Thyroid Cells

Animals were injected typically with 150 µL of the [^18^F]NaBF_4_ solution containing 55 MBq of activity. For static studies, the incorporation time was 45 min. That resulted in 41 MBq of activity at the start of the study. For a typical animal of 273 g at the age of more than two years, this was a dose of 0.64 pmol of ^18^F and 53 nmol of NaBF_4_ and, hence, boron. This is the dose of 21 µg/kg of NaBF_4_ and 2.1 µg/kg and 2.1 ppb of boron.

PET images of the thyroid region in a control animal, an animal with thyroid hyperplasia, and with tumor are shown in Figure 2 for [^18^F]NaBF_4_ and [^18^F]fluorodeoxyglucose ([^18^F]FDG) oncological tracers.

Delineating a VOI at 50% of the maximum SUV for [^18^F]NaBF_4_, we can measure the volume of the normal thyroid right lobe as 3.7 µL and tumor lobe as 40 µL. For [^18^F]FDG, neither the control nor thyroid with hyperplasia show an abnormal uptake; only the tumor shows two SUV_max_. The *xyz* dimensions of the tumor measured on the [^18^F]NaBF_4_ PET image are 4.4 × 3.8 × 4.8 mm, while the tumor extracted about two weeks later shown in Figure 2c is 7 × 5 × 5 mm, more than twice the volume from the [^18^F]NaBF_4_ PET image.

The time-activity curve for the hyperplastic thyroid as in the central part of Figure 2b is shown in Figure 3.

The maximum activity concentration is reached about 17 min after injection, reaching 2.3 MBq/mL (nine SUV_max_). After reaching the maximum, the activity concentration slowly drops to about 70% of the maximum one at 90 min when the study was finished. The *k*_3_/*k*_4_ (known as the nondisplaceable binding potential, BP_ND_) ratio is only 1.1 but is sufficient to maintain a slow activity concentration decrease. Such a slow decrease is beneficial for the multi-hour BNCT treatment.

To calculate the boron concentration, we start from calculating the ^18^F molar concentration in the product to be injected. At the end of synthesis (EOS, decay-corrected to PET study start), we received 4.3 GBq of [^18^F]NaBF_4_ in 2 mL of saline. As the activity concentration *A* is A=N/τ, where *N* is the number of ^18^F atoms and $\tau =t_{1/2}/\ln\left(2\right)$, *t*_1/2_ of ^18^F being 109.7 min expressed in seconds, and diluted in 25.9 mL of Milli-Q water, we obtained 33.9 nM of [^18^F]NaBF_4_ at the EOS and 351 µM of unlabeled NaBF_4_. We can calculate the molar concentration in the hyperplasia maximum from the maximum of the curve in Figure 3: 2.3 MBq/mL. The peak 2.3-MBq/mL activity concentration corresponds to 36.8 pM [^18^F]NaBF_4_. The ratio of concentrations of [^18^F]NaBF_4_ to NaBF_4_ is 9.7 × 10^−5^ and, with decay correction, is constant throughout the PET study. Dividing the calculated concentration of [^18^F]NaBF_4_ 36.8 pM by 9.7 × 10^−5^, we receive, therefore, a 381 nM boron concentration, which is 19 × 10^−5^ of the fraction of required 2 mM ^10^B concentration for BNCT treatment.

The result of applying two-tissue compartment modeling is shown in Table 1.

The distribution volume *Vt* is high at 7.56, while the distribution volume of the specific binding *Vs* is 3.96. The goodness-of-fit R^2^ from the Levenberg–Marquardt algorithm for this model is 0.95.

### 3.4. Simulation of Epithermal Neutrons Irradiation

The cross-section for the thermal neutrons at 25 meV for the reaction ^10^B(n,α)^7^Li is 3840 barns [16]. We assume here the best possible scenario, in which all the neutrons produced in the target are fully thermalized while reaching thyroids. This can be accomplished by tuning the right voltage of the TANDEM and designing an optimized Beam Shaper Assembly for the HISPANoS beamline. The distance in the tissue from the surface to the thyroid is known and, for a horizontal beam, is 13 mm (7 mm is the shortest distance to the skin; see Figure 2c). Considering the neutron fluence in the HISPANoS of 3·10^7^ n/cm^2^/s at 10 cm, which we assume here as a distance to the thyroids, the number of ^10^B(n,α) reactions in a normal 3.7 µL and a tumor thyroid lobe of 40 µL of a rat irradiated for 1 h are summarized in Table 2 for two concentrations of ^10^B in columns two and three.

If the total energy released in the ^10^B(n,α)^7^Li reaction, 2.3 MeV, is deposited in the thyroids, the calculated physical dose is shown in Table 1 in the fourth column. Assuming the Radiobiological Effect (RBE) of neutrons captured in ^10^B to be 4.2 [17], the effective dose is shown in the fifth column of Table 2 in Gy equivalent units.

## 4. Discussion

With the radiotracer injection, we reached 36.8 pM [^18^F]NaBF_4_ and, therefore, a 381 nM boron concentration, which is 19 × 10^−5^ of the fraction of boron required in the 2 mM concentration for BNCT treatment. The exploratory nature of the PET is not meant to reach BNCT therapeutic ^10^B concentrations. We injected about 21 µg/kg of natural NaBF_4_ in the PET study with [^18^F]NaBF_4_; so, to reach 2 mM of (^10^B) for BNCT, we need to inject a 110-mg/kg dose of ^10^B-enriched [^10^B]NaBF_4_. The required concentration in the tumor is reached in 17 min and maintained high enough to avoid continuous infusion, as it is done with BPA in BNCT [3]. There were previous attempts to calculate the boron concentration in vivo using PET, but they relied on the similarity of BPA and [^18^F]FBPA [17], where hydrogen in the phenylalanine ring is replaced by ^18^F. Due to the similarity, the [^18^F]FBPA and BPA cannot be measured in the same study or, more importantly, neither during nor just before the BNCT treatment using BPA. Watabe et al. [18] calculated the boron concentration by calibrating the PET-derived SUV_max_ results with inductively coupled plasma optical emission spectrometry. There, the difference between the calibration-derived boron concentration was from +10% to −48% and was statistically significant for the lung, intestine, and kidney.

Such a linear extrapolation to the dose of 110 mg/kg may be hindered by an inhibitory constant (IC_50_) of NIS for NaBF_4_, which is 1.6 µM [19]. The initial concentration of NaBF_4_ in the blood after injection of 110 mg/kg in the rat is about 16 mM, but as it can be seen from Figure 3, it is expected to drop fast to about 25% at 90 min—that is, to 4 mM. BPA is injected at doses of 500 mg/kg [3], which in rats, results in an initial concentration in the blood of 37 mM. The transporter of BPA is L-type amino acid transporter (LAT1), which for the most similar compound, phenylalanine has an IC_50_ of about 60 µM [20]. This is a five-to-six orders of magnitude higher concentration in the blood for phenylalanine, and BPA still allows reaching therapeutic boron concentrations in the tumor. As for the BPA IC_50_ of phenylalanine being not a limiting factor, we do not expect it to be a limiting factor for NaBF_4_.

The HISPANoS system does not deliver in 1 h of irradiation enough RBE dose to the tumor, which, typically, is about 20 Gy. However, for some studies, doses of 4 Gy are used. As follicles occupy only some fraction of the thyroid volume, boron is expected to follow the NIS spatial concentration and, therefore, effectively reach up to three times the thyroid concentration averaged over the full thyroid volume. This would increase the effective dose. With a relatively slow excretion of NaBF_4_ from the thyroid, the neutron irradiation time can be increased for the preclinical study up to 3–4 h in a heated animal bed, which we perform in our PET lab as maximum time.

## 5. Conclusions

The treatment of some thyroid cancers involves resection of the whole thyroid, followed by ^131^I ablation therapy with several GBq of activity. The consequence of the use of ^131^I is several days of isolation of the patient due to their radioactivity and to excrete the excess radioactive iodine. Women are advised not to have pregnancy following six months after such therapy. We propose the use of NaBF_4_ with BNCT instead of the ^131^I treatment, which removes the two mentioned side effects of ^131^I therapy, as there is no post-treatment radioactivity.

A boron concentration in the tumor—or, even better, its 3D map in the whole tissue on the passage of the neutron beam—is essential for BNCT treatment planning. Up to now, it has been evaluated either by the use of a similar compound in PET, as is the case for [^18^F]FBA and BPA, or by ex vivo methods or by inferring the boron concentration in the tumor from the boron concentration in the blood. In this work, the use of the same compound differing only in the fluorine isotope ratio allows a theranostic injection of a mixture of NaBF_4_ and [^18^F]NaBF_4_ shortly before the treatment. A futuristic view is to inject such a mixture, determine the boron concentration 3D map from the PET, qualify for therapy, plan the BNCT therapy, and execute it all within two h. PET image reconstruction needs a few minutes now, treatment planning needs much more time, but with the computer power increase and parallel and cloud computing, we may reach this level soon. NaBF_4_ is a compound very specific for organs with natural NIS expression (thyroids, stomach, and pituitary gland), but the ease of labeling can be applied to any compound of the R-BF_n_ type, which can sustain heating in a water solution to about 120 °C [21]. Such R-BF_n_-type compounds may be used in the future for BNCT theranostic treatments. Last, it should be remarked that the CNA where this research is being carried out features all the facilities required to carry out and test the ideas proposed herein: the radiopharmacy laboratory, the animal house, the PET scanner, and the HISPANoS neutron beam.

## Figures and Tables

**Figure 1 cells-09-02084-f001:**
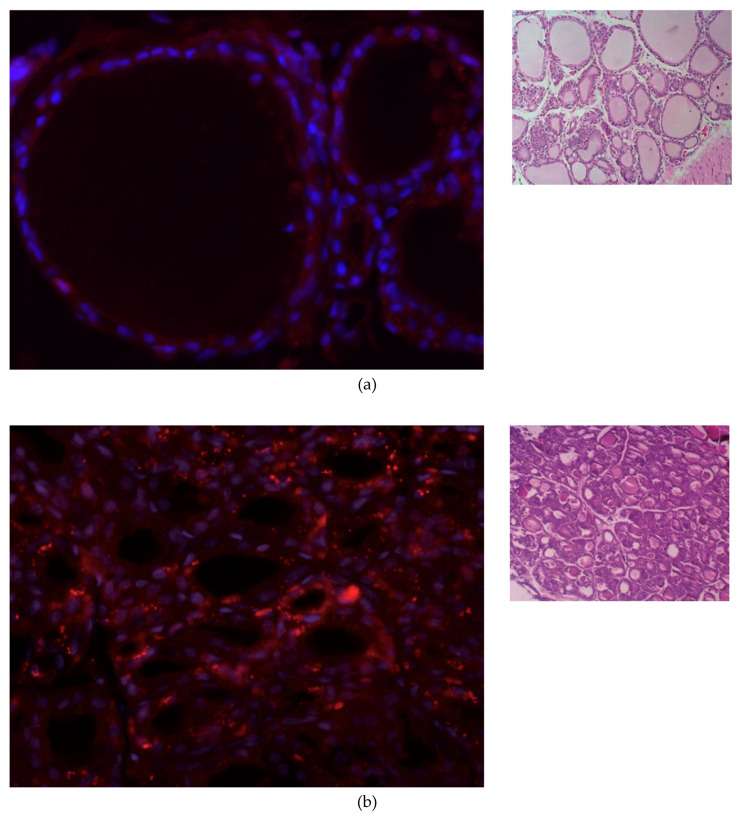
Immunofluorescence (left) and hematoxylin and eosin (H&E) staining (right) of thyroid tissue. Red: NIS primary antibody, and blue: nuclear counterstaining with DAPI. (**a**) Normal thyroid. (**b**) Thyroid with hyperplasia. (**c**) Papillary thyroid cancer. A papilla formation is observed. Immunofluorescence image magnification is 400. H&E staining magnification in (**a**) is 100 and in (**b**) and (**c**) is 200.

**Figure 2 cells-09-02084-f002:**
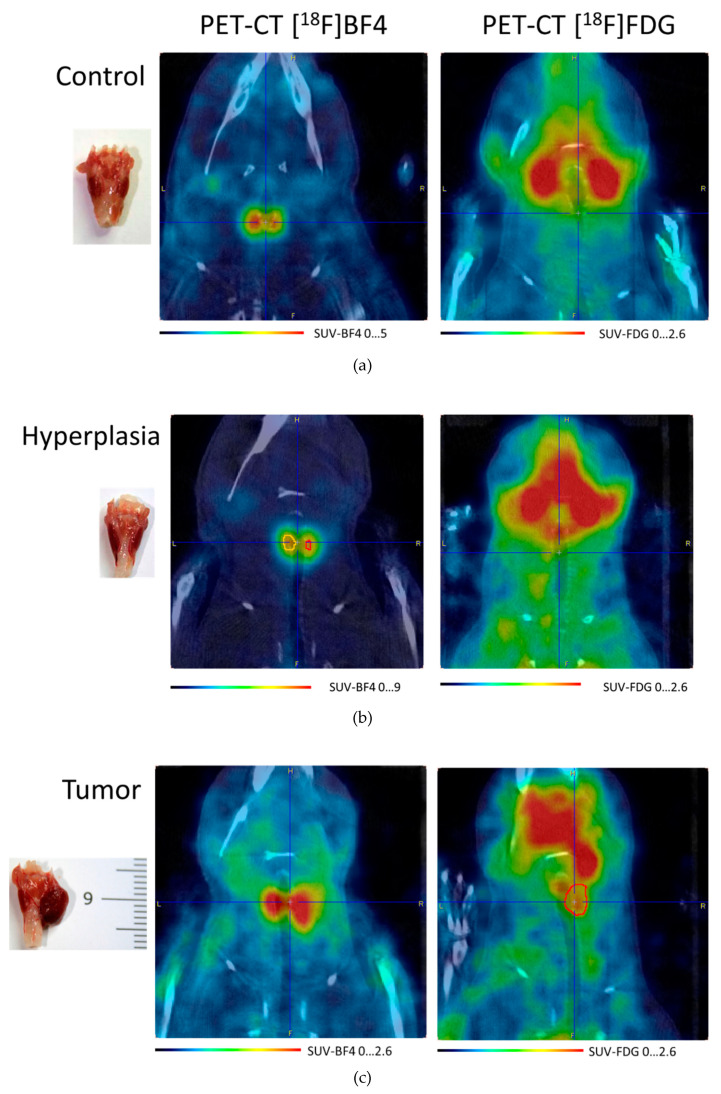
Dorsal view of the thyroid gland. PET-CT in vivo images obtained using [^18^F]NaBF_4_ ([^18^F]BF4) and [^18^F]fluorodeoxyglucose ([^18^F]FDG). Blue cross centered in the thyroid area. (**a**) Control animal. (**b**) Hyperplastic thyroid gland. The volumes of yellow and red show the thyroid gland, which appears to be the same size in the PET image, with a resolution of 2.3 mm. Pay attention to the uptake of the FDG in the annular muscle, the front to the blue cross. (**c**) Tumor in the thyroid gland. Red: volume of interest (VOI) shows the outline of the tumor for the SUV_max_ 1.7… 2.4. [^18^F]BF4) and [^18^F]FDG images were taken within eight days for each animal. The point of high metabolism of the anterior-internal FDG close to the blue cross is the annular muscle that opens the larynx. On the left, tumor scale in mm. PET/CT DICOM image files are available as Appendix A and as reference [15].

**Figure 3 cells-09-02084-f003:**
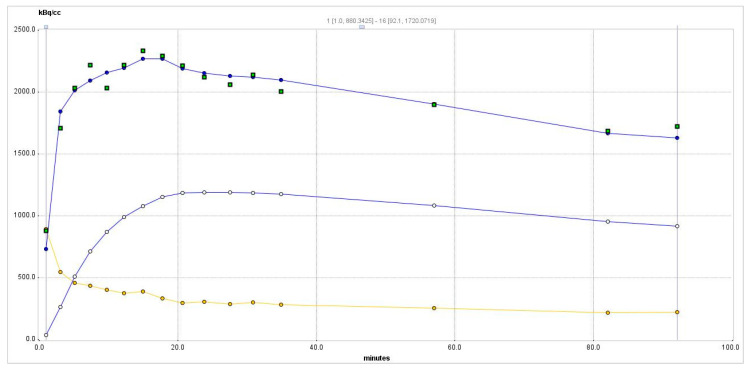
Time-activity curve of the PET radiotracer in the hyperplastic thyroid. Squares—experimental data point from the right thyroid. The blue curve passing among these points is a two-tissue compartment model curve from Equation (1). Yellow circles show whole-blood data derived from the image; the yellow curve is the whole-blood curve. Empty circles show the specific binding concentration C_2_.

**Table 1 cells-09-02084-t001:** Two-tissue compartment modeling parameters for the data shown in Figure 3 using Equation (1).

K1	k2	k3	k4	*Vs*	*Vt*	Kl/k2	k3/k4	Flux
1.89	0.52	0.09	0.08	3.96	7.56	3.60	1.10	0.28

**Table 2 cells-09-02084-t002:** Neutron HISPANoS (Hispalis Neutron Source) 1-h irradiation parameters of a rat thyroid lobe.

	Normal Thyroid	Tumor Thyroid		
Concentration	Captured n	Captured n	Dose, Gy	Dose, Gy-e
1 mM ^10^B	257	2770	0.092	0.388
2 mM ^10^B	513	5550	0.185	0.776

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
