# Peer review of "Quantification of Boron Compound Concentration for BNCT Using Positron Emission Tomography"

_cells, 2020, doi:10.3390/cells9092084_

Round 1

Reviewer 1 Report

The paper by Marcin Balcerzyk et al. investigated the use of Positron Emission Tomography imaging to quantification of 10B in thyroid tumour for Boron Neutron Capture Therapy (BNCT) to determine the role of BNCT in the treatment for thyroid cancer. There are studies performed previously to determine the boron concentration in other tumours, predominantly in glioma.

Other similar studies found:

  • Mitsuyoshi Yoshimoto, et al. (2018). Non‐invasive estimation of 10B‐4‐borono‐L‐phenylalanine‐derived boron concentration in tumors by PET using 4‐borono‐2‐18F‐fluoro‐phenylalanine. Cancer Science, 109(5), 1617-1626.
  • Tadashi Watabe et al. (2017). Practical calculation method to estimate the absolute boron concentration in tissues using 18F-FBPA PET. Annals of Nuclear Medicine, 31(6), 481-485.
  • Kohei Hanaoka, et. (2014). FBPA PET in boron neutron capture therapy for cancer: Prediction of 10B concentration in the tumor and normal tissue in a rat xenograft model. EJNMMI Research, 4(1), 1-8.

Strength of the study

  1. Well combined approach including the use of Immunofluorescence and Positron Emission Tomography to prove the hypothesis
  2. The incorporation of animal study to further support the potential of adhesamine for biological research as well as clinical application

The layout of the article was well organized and the study have shown evidence to support the potential use of BNCT for thyroid cancer. However, there are some minor concerns remain.

  1. Line 28, please specify whether the SUV measurement was in the form of SUVmean or SUVmax.
  2. Line 72, it was mentioned that the rats were 16-17 weeks old, but at line 80, it was mentioned that animals from treated group received 1% potassium perchlorate (W/V) in the drinking water, ad libitum, for 18 months, please clarify the duration of the treated group received 1% potassium perchlorate (W/V) in the drinking water.
  3. Line 105: what was the activity of the radiotracer injected into the rats?
  4. Line 110-111: what was the mA setting in the CT scan?
  5. Line 112-113: Any attenuation correction used for PET used?
  6. Line 126: is the proton beam 2.5MeV instead of 2,5 MeV?
  7. Section 2.6. Line 132 to 140. It is not clear that how the boron concentration was calculated.
  8. Line 179, the dosage of F-18 and NaBF4 for typical animal of 273g was mentioned, please correspond the dose to the weight of the animal used in this study, as it was mentioned at line 72 that rats weighted 200-220g were used.
  9. Line 185, please specify whether the SUV measurement was in the form of SUVmean or SUVmax.
  10. Line 205, please specify whether the SUV measurement was in the form of SUVmean or SUVmax.
  11. Line 205, is the activity concentration of 70%, already corrected for decay?
  12. Line 213, should be MeV instead of meV.
  13. Line 251, it is better to use the term 131I ablation therapy or 131I radiation therapy instead of brachytherapy.

Author Response

The comments of the reviewer are in italics. My answers are in normal text.

The paper by Marcin Balcerzyk et al. investigated the use of Positron Emission Tomography imaging to quantification of 10B in thyroid tumour for Boron Neutron Capture Therapy (BNCT) to determine the role of BNCT in the treatment for thyroid cancer. There are studies performed previously to determine the boron concentration in other tumours, predominantly in glioma.

Other similar studies found:

  • Mitsuyoshi Yoshimoto, et al. (2018). Non‐invasive estimation of 10B‐4‐borono‐L‐phenylalanine‐derived boron concentration in tumors by PET using 4‐borono‐2‐18F‐fluoro‐phenylalanine. Cancer Science, 109(5), 1617-1626.
  • Tadashi Watabe et al. (2017). Practical calculation method to estimate the absolute boron concentration in tissues using 18F-FBPA PET. Annals of Nuclear Medicine, 31(6), 481-485.
  • Kohei Hanaoka, et. (2014). FBPA PET in boron neutron capture therapy for cancer: Prediction of 10B concentration in the tumor and normal tissue in a rat xenograft model. EJNMMI Research, 4(1), 1-8.

The studies are now considered at the end of the first paragraph of Section 4, Discussion.

Strength of the study

  • Well combined approach including the use of Immunofluorescence and Positron Emission Tomography to prove the hypothesis
  • The incorporation of animal study to further support the potential of adhesamine for biological research as well as clinical application

 We did not use adhesamine in this study.

The layout of the article was well organized and the study have shown evidence to support the potential use of BNCT for thyroid cancer. However, there are some minor concerns remain.

  • Line 28, please specify whether the SUV measurement was in the form of SUVmean or SUVmax.

It was SUVmax. The subscript was added.

  • Line 72, it was mentioned that the rats were 16-17 weeks old, but at line 80, it was mentioned that animals from treated group received 1% potassium perchlorate (W/V) in the drinking water, ad libitum, for 18 months, please clarify the duration of the treated group received 1% potassium perchlorate (W/V) in the drinking water.

We treated the animals indeed for 18 months with the acceptance of the ethics committee. The weight was at the beginning of the study. All animals were females, so they do not grow beyond 300 g. The reference to the treatment method was added in Section 2.1.

  • Line 105: what was the activity of the radiotracer injected into the rats?

It was 55 MBq, the information was added to the sentence.

  • Line 110-111: what was the mA setting in the CT scan?

The current was 177 µA. This information was added to the CT parameters sentence at the end of the first paragraph of Section 2.4.

  • Line 112-113: Any attenuation correction used for PET used?

No, no attenuation correction was used. The appropriate information was added to the sentence.

  • Line 126: is the proton beam 2.5MeV instead of 2,5 MeV?

Corrected to 2.5 MeV.

  • Section 2.6. Line 132 to 140. It is not clear that how the boron concentration was calculated.

That was explained briefly at the end of Section 3.3. The description was added as a second paragraph to Section 2.6, part of the Methods. The particular calculation for data shown in Figure 3 is added at the end of Section 3.3, second paragraph below Figure 3.

  • Line 179, the dosage of F-18 and NaBF4 for typical animal of 273g was mentioned, please correspond the dose to the weight of the animal used in this study, as it was mentioned at line 72 that rats weighted 200-220g were used.

As I explained in point 2, the animals at the beginning of the study at 16 weeks were 200-220 grams. In that PET study, the animal (female) was done at 2 years and 3 months of age. In two years it grew 50 grams, which is typical for Wistar rat female. I added the information about the age of the animal to the sentence.

  • Line 185, please specify whether the SUV measurement was in the form of SUVmean or SUVmax.

It was SUVmax. The subscript was added.

  • Line 205, please specify whether the SUV measurement was in the form of SUVmean or SUVmax.

It was SUVmax. The subscript was added.

  • Line 205, is the activity concentration of 70%, already corrected for decay?

Yes, like all data of dynamic studies are corrected for 18F decay, as explained in Section 2.6.

  • Line 213, should be MeV instead of meV.

The energy of 25 meV comes from the kT at 300 K, k being Boltzman constant. For that energy, the thermal energy is taken for thermalized neutrons.

  • Line 251, it is better to use the term 131I ablation therapy or 131I radiation therapy instead of brachytherapy.

The phrase “brachytherapy” was changed to “ablation therapy”.

Reviewer 2 Report

The manuscript is well written (although it could benefit from undergoing extra proofreading). The authors should keep either American or British english. Unfortunately the quality and resolution of Figure 3 is not good/publishable.

It seems the authors do not use partial Volum correction (PVC) for the exact estimation of the concentration. Moreover, the reviewer is interested to know the amount of defluorination of F-18-NaBF4 (in vivo). 

The kinetic modeling is not well discussed. The authors should discuss the modeling and in vivo binding potential of NaBF4. The activity concentration in plasma (blood input function) is also related to the fate of the tracer (in vivo stability). The in vivo stability of the tracer is yet not explained.    

Author Response

The comments of the reviewer are in italics. My answers are in normal text.

The manuscript is well written (although it could benefit from undergoing extra proofreading). The authors should keep either American or British English.

As the MDPI template English is United States, I tried to keep up with that language version choice. I removed “tumour” and replaced it by tumor. I also removed several occurrences of comma as decimal separator and replaced it by point. Animal “stabulary” was replaced by animal house. Language was passed through Grammarly correction in Word add-in.

Unfortunately, the quality and resolution of Figure 3 is not good/publishable.

The axes of Figure 3 were scaled from zero to full-tens values. The numerical data were moved to Table 1.

It seems the authors do not use partial Volume correction (PVC) for the exact estimation of the concentration.

Yes, this is true. A relevant phrase was introduced in second paragraph of Section 2.4.

Moreover, the reviewer is interested to know the amount of defluorination of F-18-NaBF4 (in vivo). 

The defluorination is negligible. Already in Figure 2 in images related to NaBF4, the lower jaw does not show any signal difference from neighboring muscle. SUV there is 1.3 , while in the neighboring muscle is about 1 SUV, 0.45 SUV in the spine and 0.2 SUV in the brain, where the only flow is in the blood, as this compound does not pass blood-brain barrier. All SUV are local point measurement. The DICOM images are shown in https://doi.org/10.5281/zenodo.3968289 , which is also a Reference now.

The kinetic modelling is not well discussed. The authors should discuss the modeling and in vivo binding potential of NaBF4. The activity concentration in plasma (blood input function) is also related to the fate of the tracer (in vivo stability).

The additional explanation of kinetic modelling is added at the end of Section 2.4 and at the end of Section 3.3, after Figure 3. In vivo stability is very good as determined in Reference 5, no defluorination and no other identified metabolites in our hands. I hope that the additional information is sufficient now.

The in vivo stability of the tracer is yet not explained.

The relevant sentence is placed in Section 2.4 as a first sentence in third paragraph. This feature was studied in Reference 5 by the first users of this tracer.